# Effects of Environmental Enrichments on Welfare and Hepatic Metabolic Regulation of Broiler Chickens

**DOI:** 10.3390/ani14040557

**Published:** 2024-02-07

**Authors:** Seong W. Kang, Karen D. Christensen, Michael T. Kidd Jr., Sara K. Orlowski

**Affiliations:** 1Department of Poultry Science, Center of Excellence for Poultry Science, University of Arkansas, Fayetteville, AR 72701, USA; michaelk@uark.edu (M.T.K.J.); orlowski@uark.edu (S.K.O.); 2Tyson Foods, Inc., Springdale, AR 72762, USA; karen.christensen@tyson.com

**Keywords:** enrichment hut, variable light intensity, hepatic fatty acid, welfare, serotonin, dopamine, BDNF

## Abstract

**Simple Summary:**

Environmental enrichment (EE) has been suggested to increase environmental complexity and can benefit broilers’ welfare. In the previous lighting enrichment study in commercial broilers, a variable light intensity (VL) lighting program or gradient lighting program improved the voluntary natural behavior and activity for better health and production efficiency of broilers. In the present study, we observed the best engagement in enrichment huts (EHs) from the three types of EEs (board, ramp, and hut), implying that EHs provide a lower light intensity area which birds can rest inside for safety and mental health. The combined effect of two EEs, VL lighting and EHs, on mental health and hepatic metabolic function indicate that the enriched lighting and hut program improve the mental health and hepatic fatty acid/glucose metabolic functions, suggesting that the combined enriched lighting and hut programs make broilers sentient to light intensity in the broiler house and improve mental health and nutrient-use efficiency in the liver.

**Abstract:**

The aims of this study were to find suitable environmental enrichment (EE) and evaluate the combined effect of two EEs, variable light intensity (VL) lighting program and EH, on mental health and hepatic metabolic regulation in commercial broilers. To find the advantageous EEs for broilers, three different EEs (board, hut, and ramp) were tested in trial 1. EEs were placed and the engagement of birds to EEs, dustbathing behavior, and daily physical activity were observed. Birds treated with huts showed higher engagement than the board- or ramp-treated birds (*p* < 0.05). The results of dustbathing behavior and daily physical activity indicated that the environmental hut (EH) is the most favorable enrichment for broilers. In the second trial, to test the effect of EHs on mental health and hepatic metabolic conditions, the brain and liver were sampled from the four treatment birds (20 lx_Con, 20 lx_Hut, VL_Con and VL_Hut) on day 42. The lower expression of TPH2 (tryptophan hydroxylase 2) of VL_Hut birds than those of VL_Con and 20 lx_Hut treated birds suggests the combining effect of EHs with the VL lighting program on the central serotonergic homeostasis of broilers. Reduced expressions of TH (tyrosine hydroxylase), GR (glucocorticoid receptor), BDNF (brain-derived neurotrophic factor) of VL_Hut treated birds compared to those of VL_Con and 20 lx_Hut birds suggest lower stress, stress susceptibility, and chronic social stress in VL_Hut treated birds. The expression of CPT1A (carnitine palmitoyl transferase 1) increased over three-fold in the liver of VL_Con birds compared to 20 lx_Con birds (*p* < 0.05). EHs treatment in VL birds (VL_Hut) significantly decreased CPT1A but not in 20 lx birds (20 lx_Hut). The expression of ACCα (acetyl-CoA carboxylase alpha) was significantly decreased in VL_Con birds compared to 20 lx_Con birds. There was no significant difference in the hepatic FBPase (fructose-1,6-bisphosphatase), GR, and 11β-HSD1 (11 β-hydroxysteroid dehydrogenease-1) expression between 20 lx_Con and VL_Con birds, but EHs significantly stimulated GR in 20 lx_Hut birds, and stimulated FBPase and 11β-HSD1 expression in the VL_Hut birds compared to 20 lx_Con birds, suggesting that the VL lighting program reduced fatty acid synthesis and increased fatty acid β-oxidation in the broilers’ liver and VL_Hut improved the hepatic de novo glucose production. Taken together, the results suggest that the stimulated voluntary activity by EHs in the light-enriched broiler house improved mental health and hepatic metabolic function of broilers and may indicate that the improved hepatic metabolic function contributes to efficient nutritional support for broilers.

## 1. Introduction

Intensification systems in broiler production to produce more efficient meat products have many benefits including improved biosecurity and environmental control, but have a negative effect on animal stress and welfare by preventing animals from showing innate behaviors that they would perform in their natural environment [1,2]. Therefore, finding novel methods to reduce the stress, and improve welfare and natural behaviors associated with barren broiler houses on-farm are critically needed. Animal environmental enrichments (EEs) can be defined as any safe materials or objects added to the animal environment which engage the brain and encourage the expression of natural behavior and exercise [3,4].

Environmental light, especially light intensity, is an important component that can exert influence on behaviors, welfare, and production for intensively housed broilers [5,6,7]. In previous light preference studies, birds showed proclivity for the brighter light area when they are presenting active behaviors but for darker light areas when resting [8,9]. In an enriched light study in commercial broiler houses, which provided LED light over the feedlines and adjusted dimmer light areas on the side wall and middle of houses, birds showed enlarged natural behaviors and daily activity, which improved footpad condition and leg health of birds [10]. However, knowledge of the effects of other environmental enrichments on broilers’ behavior and other welfare indicators including mental and hepatic functions is still limited [11].

Animal welfare is the outcome of physical and psychological well-being, which is not a continual state but rather the result of certain mental changes underlying innate motivated behaviors and learned responses [12]. The neurobiological system related to food seeking is perhaps one of the most critical components for survival and generates the attention and motivation that helps animals to find, detain, and acquire the object of their want [12,13]. The food seeking and reward system consists mostly of dopaminergic neurons in the midbrain ventral tegmental area (VTA) [14,15]. In recent studies, serotonin (5-HT), dopamine (DA), and brain-derived neurotropic factor (BDNF) were suggested in the assessment of animal welfare as positive indicators in the VTA [10,16,17]. BDNF is a stress- and physical-movement-dependent neurotrophic factor associated with emotional and cognitive function and was suggested as an animal health indicator of environmental enrichment (EE) in domestic swine [17,18]. Physical activity is essential to the augmented expression of BDNF in the brain and may enhance memory performance and reduce depressive symptoms by promoting neurogenesis and neuronal differentiation [19,20]. Glucocorticoids (GCs, corticosterone in birds) are glucoregulatory hormones that are synthesized in response to stimuli including stress and are regularly used in the assessment of animal welfare [21]. Their action depends on the GC receptor (GR) which translocate into the nucleus upon ligand binding and regulates transcription of responsive genes. VTA-GR is well known for regulating the stress and reward system regulating feeding behavior [22,23]. In recent studies of broiler chickens, avian VTA was suggested as the important region, consisting of a variety of neurons of the avian midbrain, that is involved in light perception by Opn4 (melanopsin) and might be involved in broilers’ welfare [10,24].

Physical exercise and play have a significant beneficial impact on brain and liver health in mammalian species including humans [25,26,27]. The liver is the central metabolic organ in controlling metabolic homeostasis, acting as the primary site for lipid metabolism, where more than 90% of de novo fatty acids are synthesized in chickens [28,29]. Under healthy conditions, the lipid metabolism is regulated in the liver to meet systemic energy needs in the fed and fasted states by the tightly regulated processes of fatty acid uptake, synthesis, and oxidation pathways [30]. When one or more of these processes are abnormally regulated, excess lipid accumulation can occur [30]. Hepatic lipid metabolism closely interacts with glucose metabolism in liver diseases [31,32]. The liver stores glucose in the form of glycogen and releases glucose into circulation by either glycogenolysis or gluconeogenesis [33]. In the feed state, hepatic glucose production is suppressed by insulin secretion, and the glucose ingested is stored in part as glycogen [34]. Gluconeogenesis is an intricate process that requires several enzymatic steps, which are under the regulation of hormones, nutrient intake, stress conditions, and substrate concentrations. Fructose -1,6-bisphosphatase (FBPase) is a key enzyme of gluconeogenesis, and its deficiency causes hypoglycemia and fatty liver disease [35]. FBPase deficiency should be considered as an etiology of hepatic steatosis [35]. Increased levels of circulating GC levels have been associated with pathological conditions characterized by fatty liver disease [36,37].

Therefore, addressing the midbrain-VTA-welfare-related genes and hepatic metabolic regulating genes may provide critical data to understand possible adaptive physiological responses of broilers to environment enrichments which are involved in the mental and hepatic welfare of broilers. In the present study, we hypothesized that when broilers in commercial houses are provided appropriate environmental enrichments, it will stimulate birds’ innate natural behavior, and consequently, cause voluntary movement for the consumption of feed and water, and improve physical activity and mental and hepatic welfare of birds.

## 2. Materials and Methods

### 2.1. Experimental Design and Animal Housing

Trial 1 was performed to test the engagement of birds to different enrichments (board, hut, and ramp), and the effects of the selected enrichment on the daily physical activity, natural behavior, and dustbathing. Day-old broilers (Cobb 700, mixed sex, 19,200 birds/house, stocking density_12.3 birds/m^2^) were housed in four commercial broiler houses (Tyson Foods Broiler Welfare Research Farm (BWRF)). Two replicate trials were performed, and each house was composed of 4 quadrant sections. Each quadrant of the house was placed with 4800 chicks with all source flocks equally represented in each quadrant. Each house was equipped with standard feeders, waterers, and brooders (12.8 m × 122 m, wood shavings). Two different light intensity lighting programs (20 lux (lx) and variable light intensity (VL)) were installed, and the light intensity (LED) was measured in 9 different areas of the house. The averages of light intensity in the 20 lx and VL house were 26.16 ± 0.70 lx, and 2.07/40.4 ± 0.04 lx, respectively [10]. Diet was formulated to meet minimum industry standards [38]. The light was on from 6 am day 1–3 (23 h light (L):1 h dark (D) (23 L:1 D) _40 lx); then on day 4–7, the photoperiod schedule was changed into 20 L:4 D_20 lx. On day 7, lighting programs were started for 20 lx and VL houses (16 L:8 D, light on 6 am). Environmental enrichments (EEs: board, hut, or ramp) were placed in each section of the houses (3 EEs/92.9 m^2^ (1000 sqft), 9 EEs/section).

After the enrichment hut (EH) was selected as an appropriate EE in trial 1, trial 2 was performed to assess the effects of EHs on mental health and liver metabolic functions. Day-old broilers (Cobb 700, mixed sex, 19,200 birds/house) were housed in four broiler houses (Tyson Foods BWRF). Four replicate trials were performed (*n* = 4 sections/house, 4800 birds/section). Two different light intensity lighting programs (20 lx, and VL) and the same diet were installed as in trial 1. Four treatments were assigned to each house (20 lx_Con, 20 lx_Hut, VL_Con, and VL_Hut). EHs were placed at 4 sections of 20 lx_Hut and VL_Hut house (3 huts/92.9 m^2^ (1000 sqft), 9 huts/section). At 42 days of age, birds were selected randomly in each section (*n* = 16/treatment, male), and transported to the sampling room for brain and liver sampling. The care and experimental use of animals were followed and maintained by the protocol of Tyson Foods BWRF.

### 2.2. Behavioral Observations Affected by Enrichments

#### 2.2.1. Number of Engaged Birds to Enrichments

In trial 1, a trained observer placed a camera 5 m from an enrichment within a section at the age of 28, 35, and 42 days. The observer then walked the ground adjacent to the enrichment to remove any birds actively using the enrichment. The enrichment was then recorded for 15 min. These data represent the number of birds near, under, and over the enrichment for 15 min (Figure 1D–F).

#### 2.2.2. Dustbathing and Daily Physical Activity

In trial 1, the number of dustbathing holes was counted within nine identified areas of the section (four sections/house) to see the combined effects of the two different light intensity lighting programs (20 lx and VL) and three EEs (board, hut, and ramp) on dustbathing behavior. In each section, dustbathing holes as evidence of dustbathing behavior were counted at the age of 8, 15, and 22 days [10]. The number of holes per 5 m^2^ was determined. Data were compared among treatments. In trial 1, daily physical activity was monitored using a 22 g activity tracker, Animo (www.surepetcare.com (accessed on 10 September 2023), activity and behavior monitor) which monitors animals’ activity and behavior including sleep quality, energy burnt, and shaking via tri-axial accelerometer technology [10]. A similar animal activity tracker, Fitbark (www.fitbark.com (accessed on 1 October 2023)), was used for monitoring animals’ movement in the behavior study [39]. At 43 days of age, birds were randomly selected, and body weight was measured (n = 8 birds/trt, 4 birds/section, 2 VL houses). An Animo was installed for each bird using a commercially available chicken harness and uninstalled at 48 days of age. Average daily activity (joules/day, 4 days) of each bird was obtained from the installed software. Animo energy calculation is based on an industry standard calculation that considers the bird’s weight. The energy burnt is tracked against each movement type for a bird.

### 2.3. Dissection of Ventral Tegmental Area (VTA) of Midbrain and Liver Sampling

According to previous studies in avian species and a chick brain atlas [8,10,40], the VTA regions from the sampled birds were dissected in cryostat microtome. Dissected section dimensions were 3–3.5 mm (W) × 2–3 mm (H) × 1–1.2 mm (L) for VTA. The thickness (W, H, and L) of the dissected brain tissue block was proportionally increased for 42 days’ birds based on brain size and structure. Inside the cryostat, brain areas shown as rectangles were dissected from each flattened brain section using a scalpel handle and blade (#11) and were quickly transferred to the Trizol and then stored at −80 °C until total RNA extraction. The liver tissues from the sampled birds at 42 days of age were dissected, snap-frozen, and stored at −80 °C for total RNA extraction.

### 2.4. RNA Isolation and Two-Step Real-Time Quantitative RT-PCR

Total RNA was extracted from dissected frozen brain tissue and liver tissues using TRIzol^®^ reagent (Invitrogen Life Technologies, Palo Alto, CA, USA) followed by DNase I treatment and purification of total RNA by the RNeasy mini kit (Qiagen, Valencia, CA, USA). The RNA quality and quantity were determined using agarose gel electrophoresis and NanoDrop 1000 (Thermo Scientific, Wilmington, DE, USA). Two µg of total RNA from sectioned VTA and liver tissue were converted into cDNA with oligo (dT)_16_ primer and SuperScript IV reverse transcriptase (Invitrogen, Grand Island, NY, USA), as previously described [8,10]. The specific oligonucleotide primers were designed using the PRIMERS3 program (http://primer3.ut.ee (accessed on 15 February 2023)). Primer sets for chicken TPH2 (tryptophan hydroxylase 2), TH (tyrosine hydroxylase), GR (glucocorticoid receptor), BDNF, Opn4, CPT1A (carnitine palmitoyl transferase 1), ACCα (acetyl-CoA carboxylase alpha), FBPase, GR, and 11β-HSD1 (11 β-hydroxysteroid dehydrogenease-1) were designed, and conventional RT-PCR performed for optimizing annealing temperature for each primer set (Table 1). The PCR products were analyzed by using agarose gel electrophoresis (3%). Melting curve analysis and PCR efficiency for each selected primer set were validated with the default settings on the ABI 7500 system (Applied Biosystems LLC, Foster, CA, USA). The efficiency of PCR was evaluated by performing a dilution series experiment and the slope of the standard curve was translated into an efficiency value. Efficiency of the PCR within 95–100% was accepted for this study. A portion of the cDNA was subjected to quantitative real-time PCR (qRT-PCR) using an ABI 7500 system with Power SYBR Green PCR Master Mix (Invitrogen, Grand Island, NY, USA). Chicken glyceraldehyde 3-phosphate dehydrogenase (GAPDH), β-actin, and 18S were used as internal controls. Dissociation curves were constructed at the end of amplification for validating the quality of the data. All qRT-PCR experiments were performed in triplicate, and the values of the average cycle threshold (Ct) were determined. Delta-Ct scores for gene transcripts in each sample were normalized using Delta-Ct scores for GAPDH/β-actin/18S and expressed as the relative fold change in gene expression using the equation, 2^−ΔΔCt^. The gene name, NCBI accession numbers, primer sequences, PCR product size, and annealing temperatures used in the present study are shown in Table 1.

### 2.5. Statistical Analyses

Statistical analyses were performed using JMP^®^ 14.0 (SAS Institute Inc., Cary, NC, USA). A normal distribution was first tested by the Shapiro–Wilk test and differences among the groups were analyzed using one-way analysis of variance (ANOVA) followed by mean comparison using the Tukey’s HSD test at a significance level of *p* < 0.05. Multiple comparisons of group mean by Tukey’s HSD test were used to evaluate behavior data including engagement, dustbathing holes, and daily physical activity, and relative changes of gene expression among treatment groups for each gene. Data are presented as the mean ± SEM. A probability level of *p* < 0.05 was considered as statistically significant.

## 3. Results

### 3.1. Effects of Environmental Enrichments on Engagement, Dustbathing, and Daily Physical Activity of Broilers

The shape and dimension of the three enrichments (board, hut, and ramp) used in this study are presented in Figure 1A–C. The engagement observed includes climbing or jumping on the objects, sitting next to, and hiding underneath them. On day 28, the first testing day of engagement, the engagement to the hut was the highest in VL houses among treatments and significantly higher compared to 20 lx houses (Figure 1D). On days 35 and 42, similar patterns of engagement were observed (Figure 1E,F). There was no significant difference of engagement between the board and ramp in the 20 lx and VL treated houses. EHs significantly induced the highest engagement of birds compared to boards and ramps on days 28, 35, and 42 of age (*p* < 0.05).

Dustbathing behavior made holes in the floor of the commercial broiler houses [10]. Weekly counting of dustbathing holes in each section of the house was performed and compared among treatments (Figure 2). On days 8, 15 and 22, the numbers of dustbathing holes in 20 lx houses with different enrichments were not different among treatments, excepting the difference between 20 lx_Con and 20 lx_Board on day 15. In the VL houses, VL_Hut sections have the highest number of dustbathing holes. On day 22, the number of dustbathing holes in the VL_Hut section was the highest compared to other treatments including the VL_Con section (*p* < 0.05).

To select the best enrichment for the broilers’ activity in the VL houses, birds of each section in two VL houses were selected and were installed with Animo (activity tracker) on day 43 and uninstalled on day 48 (*n* = 4/section, 2 houses, total birds *n* = 32). Average daily consumed energy (Joule/day) by moving activity was obtained for four days (from day 44 to day 47) (Figure 3). VL_Hut birds burned 33% more energy compared to VL_Con birds (*p* > 0.05) and 66% more energy compared to VL_Board birds for their moving activity (*p* < 0.05).

### 3.2. Effects of Light Programs and EHs on Regulation of Welfare Marker Genes in the Ventral Tegmental Area (VTA) in Commercial Broiler House

To investigate the long-term effects of enrichment huts in the 20 lx and VL houses on the previously identified broiler welfare marker genes in the VTA (Figure 4), expressional changes of TPH2, TH, GR, BDNF, and Opn4 genes were determined in the VTA of broilers’ midbrain at 42 days of age (Figure 5 and Figure 6F). Expression of TPH2, an indicator of serotonergic activity in the VTA, in 20 lx birds was downregulated by EHs, and the decreased TPH2 expression in the VL birds compared to 20 lx control birds was further downregulated by EHs (*p* < 0.05). TH expression, an indicator of DAergic activity, was the highest in 20 lx birds (20 lx_Con) among other birds (*p* < 0.05), and there were significant decreases in TH expression in 20 lx_Hut and VL_Con birds (*p* < 0.05). In the VL birds, EHs affected the downregulation of TH expression compared to VL_Con birds (*p* < 0.05). Both VTA-GR expressions, a stress response indicator in VTA, in 20 lx and VL birds were strongly downregulated by EHs treatment, indicating the lower stress status in EHs-treated birds compared to 20 lx and VL control birds (*p* < 0.05). There was no difference in VTA-BDNF expression, a social stress indicator in VTA, in 20 lx birds by EHs treatment (*p* > 0.05), but EHs decreased VTA-BDNF expression significantly in the VL birds. VTA-Opn4 expression was affected by the enriched lighting program as we observed previously [10]. EHs further decreased VTA-Opn4 expression of VL birds (*p* < 0.05).

### 3.3. Combined Effects of Light Programs and EHs on Regulation of Hepatic Metabolic Pathway and Stress Response Genes

To investigate the effects of EHs in the 20 lx and VL houses on the lipid and glucose metabolic and stress response functions in liver (L), expressional changes of L-CPT1A, L-ACCα, L-FBPase, L-GR and L-11β-HSD1 were determined in the liver at 42 days of age (Figure 6). Expression of L-CPT1A, a rate-limiting enzyme of fatty acid β-oxidation in the liver, in 20 lx birds was not affected by EHs treatment, but the expression of L-CPT1A was increased about 3-fold compared to 20 lx in the VL birds.

Interestingly, L-CPT1A expression was decreased in VL_Hut birds compared to VL_Con birds (*p* < 0.05). L-ACCα expression, a rate-limiting enzyme of de novo fatty acid synthesis, was downregulated in VL_Con birds compared to 20 lx_Con birds (*p* < 0.05). EHs decreased the expression of L-ACCα in 20 lx birds, which was slightly increased in VL birds (*p* < 0.05). There was no significant difference in the expression of L-FBPase, a key enzyme of gluconeogenesis, between 20 lx_Con and VL_Con birds. Both L-FBPase expressions in 20 lx and VL birds were upregulated by EHs treatment, and L-FBPase expression was significantly higher in VL_Hut birds compared to VL_Con birds (*p* < 0.05). There was no difference in VTA-GR expression, a social stress indicator in VTA, in between 20 lx_Con and VL_Con birds. EHs increased L-GR expression in 20 lx_Hut birds compared to 20 lx_Con birds (*p* < 0.05). There was no effect of EHs in L-11β-HSD1 expression in 20 lx birds (*p* > 0.05). EHs increased expression of L-11β-HSD1 in the VL_Hut birds compared to VL_Con birds. The avian VTA was suggested as an important area of the midbrain of birds involved in the light perception by Opn4 and might be involved in the welfare of birds [8,10,24]. The expression of the Opn4 gene was determined to investigate the effects of EHs in the 20 lx and VL houses. Opn4 expression was downregulated in VL_Con birds compared to 20 lx_Con birds (*p* < 0.05), and Opn4 expression in VL_Hut birds was significantly lower than that in VL_Con birds (*p* < 0.05).

## 4. Discussion

The present study was aimed to investigate the optimal environmental enrichments with the enriched lighting program (VL lighting program or gradient lighting program), which may synergically improve the birds’ natural behavior, and physiological responses of birds. Environments are perceived by birds as either frightening to survival or to homeostatic interference, leading to behavioral responses and physiological impacts on birds [10]. A lack of environmental complexity in the broiler houses can lead to low levels of activity and the frustration of highly motivated behavior, such as dustbathing [41]. Different enrichment programs in commercial broiler houses may influence diverse behavioral and physiological impacts on birds to improve welfare. Therefore, it is important to determine enrichment(s) that the animals will actively engage with, and several effective enrichment objects for broiler chickens have been suggested including ramps, platforms, and gradient lighting [42]. An enriched lighting program (VL_Con) was effective in improving birds’ voluntary natural activity and engagement compared to 20 lx program (20 lx_Con) in this study and the previous study [10], indicating the improvement of comfort natural behavior using the VL lighting program. Broiler performance including daily weight gain and feed conversion ratio was improved, and expressional changes of mental-health-indicating genes suggested better mental health through the enriched lighting program [10]. From the tested EEs in the present study, EHs showed the highest engagement, implying that EHs provide a lower light intensity area that birds can rest inside for safety and mental health compared to control (no enrichment), board, and ramp. Chicken house conditions that do not allow dustbathing behavior of birds were suggested to cause stress, and nonperformance of dustbathing behavior has been suggested to causee the experience of stress [41]. In the present study, results of dustbathing holes from the dustbathing activity of birds were counted in three weekly observations, indicating that EHs in the VL house had a most stimulating effect on dustbathing behavior with the increasing significance of results as the birds got older. These results may suggest that EHs might significantly decrease the stress level of birds in VL birds compared to control (no EE), board, and ramp, but not significantly in 20 lx houses. An activity tracker, Animo, was used to observe daily physical activity of birds at late stage of age (Days 44–47). The daily physical activities showed a significant difference in average daily activity between VL-board birds and VL-hut/ramp birds (Figure 3), suggesting the significant role of huts and ramps for birds’ behavior and mental welfare. Environmental light has been recognized as a critical component that can affect animal behavior, and fear is an adaptive emotional and behavior response to potentially harmful stimuli and serves to defend animals. In the present study, it appears that birds engaged huts and ramps for their emotional and behavior responses in the broiler houses that provided different light intensity areas by VL lighting program and enrichment huts/ramps. Interestingly, the energy used by VL-ramp birds was not significantly different from VL-hut birds, indicating that increasing physical activity may not be correlated with dustbathing behavior in broilers. Accordingly, the results of engagement, dustbathing behavior, and daily physical activity studies indicated that EHs would be a preferable EE in the enriched lighting program broiler houses. It may be possible that the behavioral changes of broilers by the VL lighting program and enrichment hut/ramp in the present study is the epigenetic adaptation of broilers’ physiology with environmental light intensity. The Opn4 (melanopsin) expressional changes in the VTA of broilers midbrain, in the present study, indicate the possibility of light perception and integration of behavior and physiological responses by Opn4 [24].

The high-quality health of animals is not simply the absence of negative experiences, but rather it is inherently the presence of positive emotions and experiences [16,43,44]. Studies reported that the midbrain VTA contains cell bodies of mesolimbic DAergic neurons as well as the 5-HTergic system in mammals and avian species, and is associated with mental health [8,10,45,46]. To address the mental welfare of broilers under different lighting programs and EHs, we measured TPH2 and TH expressions in the VTA as an indicator of 5-HTergic and DAergic activities. Birds in the VL houses took a rest and slept in the dark-lighting area of the house and actively fed and drank water in the bright feedline area, which may contribute a favorable environment condition. Apparently, providing enrichment huts added more comfortable resting areas for birds to take a break in and around as we observed in the engagement study. Results that EHs affected downregulation of TPH2 and TH expression in both 20 lx house and VL house, suggest that the lower synthesis of these two positive welfare indicators in EHs-treated birds indicates the lower stress-susceptibility in the VL_Hut birds [8,10]. The current study used the GR expression in VTA (VTA-GR) to investigate the effect of two different enrichments (enriched lighting and huts) on stress susceptibility in the broiler brain. Downregulated GR expressions in both 20 lx- and VL-treated birds by EHs indicate that the effect of EHs on stress susceptibility occurred in both houses (20 lx and VL houses), but not specific to the lighting conditions of broiler houses. Enhancing physical activity by EE has been suggested to be a crucial strategy to improve broilers’ welfare including emotion and leg health [47,48,49]. Several brain mechanisms may explain the positive impact of increased activity, including an increase in beneficial neurotropic factors. The stimulation of neurotropic factor BDNF in VTA was suggested to be involved in the long-term social defeat stress, and the deletion of the BDNF gene in the VTA diminished stress-induced behaviors, such as social avoidance in rodents [50,51]. In the present study, downregulation of VTA-BDNF expression occurred only in the VL houses, and these results indicate that the effect of enrichment huts on social stress is specific to the VL house lighting condition. Opn4 has been linked to several behavioral responses to light, including circadian photo-entrainment, light repression of movement in nocturnal animals, and activity in diurnal animals [52]. Avian midbrain VTA was suggested as an important brain region involved in light perception and has been suggested to be associated with avian welfare [8,10]. The result of synergistic downregulation of Opn4 expression by EHs suggests that the VTA-Opn4 might be involved in the direct perception of light information for sensual adaptation and welfare of broilers.

The metabolism of lipids has some essential differences in the biosynthesis and transport of lipids between avians and mammals [53]. In avian species, the liver is the main site of lipid biosynthesis and responsible for 90% of the free fatty acids synthesized de novo [54]. Fatty liver syndrome (FLS) is an augmented hepatic triacylglycerol content and a nutritional disease caused by a metabolic disorder which has effected huge economical losses to the poultry industry, but the pathogenesis remains incompletely elucidated [55,56]. Many factors were suggested as possible causes of FLS in chickens including feed ingredient quality and inappropriate feed formulation, but imbalanced hepatic lipid metabolism may be a critical one [57]. CPT1A is a rate-limiting enzyme for fatty acid β-oxidation that catalyzes the transfer of the long-chain acyl group in acyl-CoA ester to carnitine, enabling fatty acids to enter the mitochondrial matrix for oxidation [58,59]. The deficiency of CPT1A or abnormal regulation can result in diseases like metabolic disorders [60]. Elevated hepatic CPT1A expression in VL birds compared to 20 lx birds indicates the beneficial effect of voluntary activity induced by the enriched lighting program on hepatic fatty acid β-oxidation, and the lower expression of CPT1A in 20 lx houses may be associated with the FLS of broilers. Intriguingly, this upregulated CPT1A expression by the VL lighting program was slightly downregulated by EHs, but still higher than 20 lx-Hut birds. ACCα, a rate-limiting enzyme for de novo fatty acid synthesis, determines the level of hepatic lipid content [61]. In chickens, ACCα expression was significantly upregulated in the FLS chicken, suggesting that altered de novo lipogenesis may be the main pathway of pathogenesis of FLS in chickens [55]. Downregulated ACCα expression in the VL_Con birds compared to 20 lx_Con indicated that fatty acid de novo synthesis decreased using the VL lighting program, but there was no difference in ACCα expression levels between 20 lx_Hut and VL_Hut birds, suggesting that the enriched lighting program is more critical to decreasing de novo fatty acid synthesis in broiler liver than EHs. The deficiency of FBPase, a key enzyme of de novo glucose synthesis (gluconeogenesis), is associated with hepatic steatosis, causing hypoglycemia and fatty liver disease, displaying distinct levels of glucose-derived de novo lipogenesis. [35,61]. Results in this study suggest that the upregulation of FBPase expression in the EHs-treated birds in VL houses (VL_Hut) may contribute to the improvement of hepatic welfare by efficient new glucose production from glycerol in the liver [33,62]. GR and 11β-HSD1 intermediate the regulation of intracellular corticosterone, indicating that they play fundamental roles in the pathogenesis of the metabolic syndrome [63]. Stress changes GR expression in wild birds and broilers [64,65]. The effects of EHs on the GR expression in the broiler liver occurred only in 20 lx birds, suggesting the specificity of the enrichment hut effect on GR expression in the non-light-enriched house. The regeneration of active GCs in cells by 11β-HSD1 is critical to developing the tissue specific phenotype of GC excess [66]. In fact, liver-specific overexpression of 11β-HSD1 increased hepatic lipid synthesis through an upregulation of fatty acid synthase and reduced lipid clearance in mice liver [67]. We observed hepatic 11β-HSD1 expression increased only in the VL_Hut birds that also have the increased expression of de novo fatty acid synthesis gene (ACCα) compared to control birds (VL_Con), suggesting that EHs may be involved in the upregulation of fatty acid synthesis and increase lipid removal in liver.

## 5. Conclusions

In conclusion, the results suggest that the combined enriched lighting and enrichment hut programs in broiler houses stimulated voluntary activity, and enhanced mental health and hepatic metabolic function of broilers, indicating that the improved hepatic metabolic function may contribute to efficient nutritional support for broilers by EHs. Improved hepatic lipid metabolic functions by the VL lighting program may be associated with the better performance (higher daily weight gain and lower feed conversion ratio), and the combined effects with EHs appear to be synergistic in welfare and mental health.

## Figures and Tables

**Figure 1 animals-14-00557-f001:**
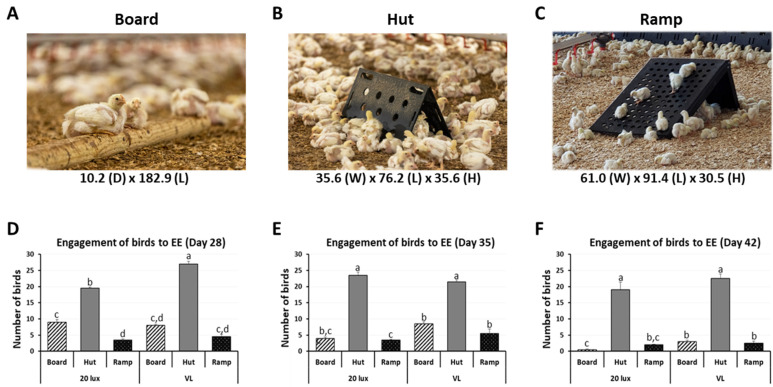
Effects of the three different environment enrichments (board, hut, and ramp) on engagement of birds at different ages. (**A**–**C**) Three commercially available enrichments placed in trial 1, 20 lx and VL houses. Enrichment dimensions: Board; 10.2 cm (diameter) × 182.9 cm (long), Hut; 35.6 cm (wide) × 76.2 cm (long) × 35.6 cm (height), Ramp; 61.0 cm (wide) × 91.4 cm (long) × 30.5 cm (height). (**D**–**F**) Weekly number of engaged birds was observed on day 28, 35, and 42 for each enrichment. Data (mean ± SEM) were compared among treatments. Different lower-case letters above the bars denote significant differences (*p* < 0.05) among groups.

**Figure 2 animals-14-00557-f002:**
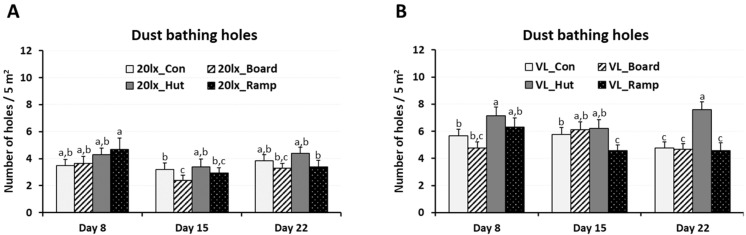
Effects of the three different enrichments (board, hut, and ramp) on the number of dustbathing holes in different light condition and ages. In each section of trial 1, 20 lx houses (**A**) and VL houses (**B**), dustbathing holes as the evidence of dustbathing behavior were counted at 8, 15, and 22 days of age. Dustbathing holes were observed in nine parts of each section and the number of holes per 5 m^2^ was determined. Data (mean ± SEM) were compared among treatments. Different lower-case letters above the bars denote significant differences (*p* < 0.05) among groups.

**Figure 3 animals-14-00557-f003:**
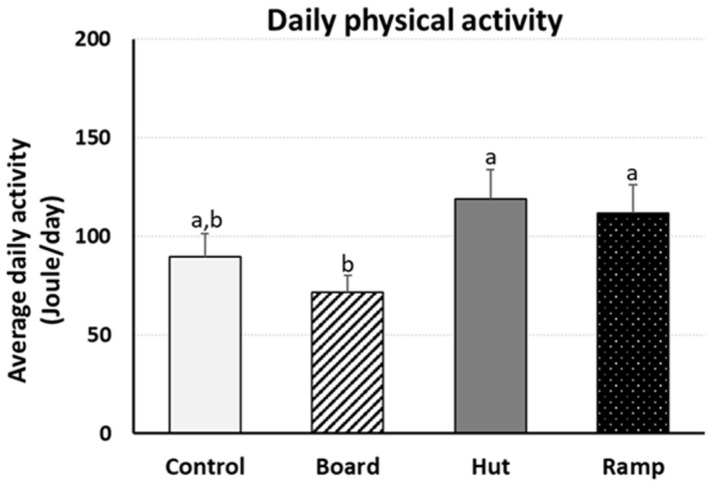
Effects of the different environmental enrichments (board, hut, and ramp) on daily physical activity. In each section of trial 1 VL houses, an activity tracker, Animo, was installed on birds using harness at 43 days of age and uninstalled at 48 days of age (*n* = 8 birds/trt, 4 birds/section, 2 VL houses). Average daily activity (calorie consumption) for each bird from day 44 to day 47 was obtained. Data (mean ± SEM) were compared among treatments (control, board, hut, and ramp). Different lower-case letters above the bars denote significant differences (*p* < 0.05) among groups.

**Figure 4 animals-14-00557-f004:**
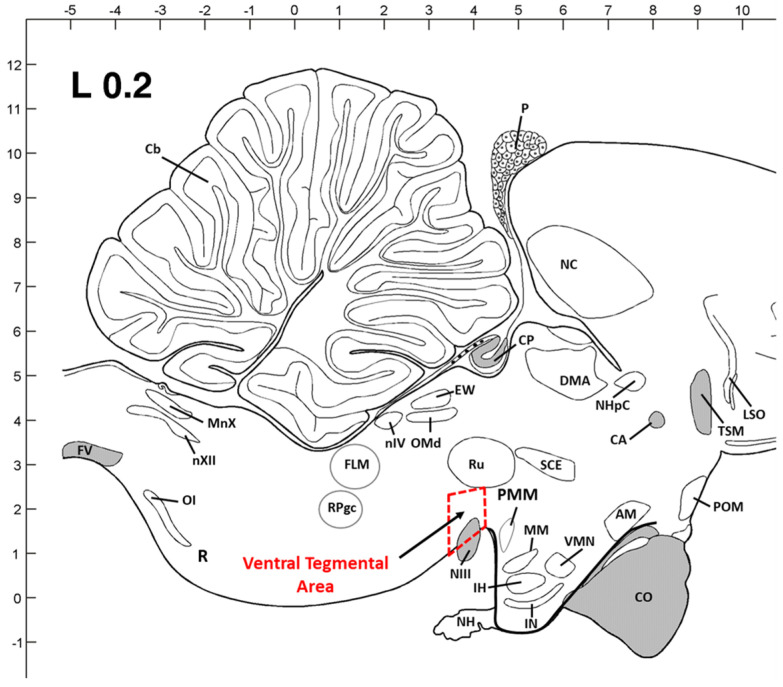
Sagittal view of dissection area of the chicken midbrain (Ventral Tegmental Area). Dimensions of dissected tissue are coronally 3–3.5 mm (W) × 2–3 mm (H) × 1–1.2 mm (L) for VTA. The thickness (W, H, and L) was adjusted proportionally from young birds to older birds based on brain size and structure. Abbreviations used: AM: anterior medial hypothalamic nucleus; CA: anterior commissure; Cb: cerebellum; CO: optic chiasma; CP: posterior commissure; DMA: dorsomedial nucleus; EW: Edinger–Westphal nucleus; FLM: medial longitudinal fasciculus; FV: ventral fasciculus; IH: inferior hypothalamic nucleus; IN: infundibular hypothalamic nucleus; LSO: lateral septal organ; MM: medial mammillary nucleus; MnX: nucleus motorius dorsalis nervi vagi; NC: caudal neostriatum; NH: neurohypophysis; NHpC: nucleus of the hippocampal commissure; NIII: oculomotor nerve; nIV: trochlear nerve nucleus; nXII: hypoglossal nerve nucleus; OI: inferior olivary nucleus; OMd: dorsal oculomotor nucleus; P: pineal gland; POM: medial preoptic nucleus; PVN: paraventricular nucleus; RPgc: nucleus of caudal pontine reticular gigantocellular; Ru: red nucleus; SCE: stratum cellular externum; TSM: septopallio-mesencephalic tract; Top VMN: ventromedial hypothalamic nucleus.

**Figure 5 animals-14-00557-f005:**
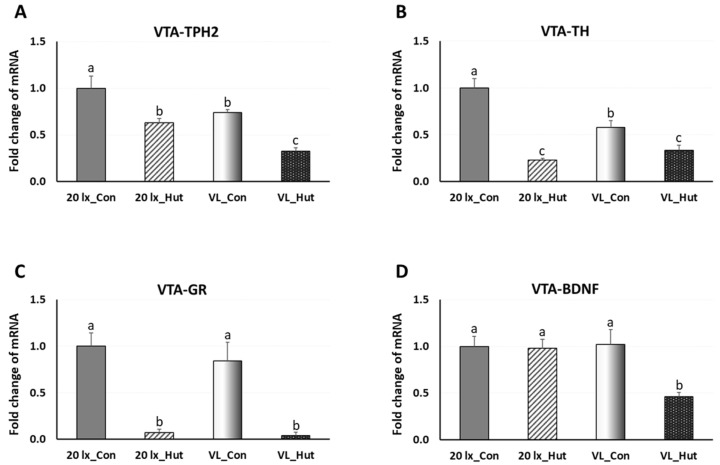
Expression changes of (**A**) TPH2, (**B**) TH (the rate-limiting enzyme of dopamine biosynthesis), (**C**) glucocorticoid receptor (GR), and (**D**) brain-derived neurotropic factor (BDNF) mRNA in the ventral tegmental area (VTA) of birds at 42 days of age. Brains of birds were sampled on day 42 (*n* = 12/section, 4 sections/house). The VTA of the brainstem from each bird was dissected as described in Materials and Methods. Total RNA was extracted and used for RT-qPCR. Data were set as the relative fold changes of expression levels using the ΔΔCt method with GAPDH and β-actin as internal controls. Data (mean ± SEM) were expressed from a value set for 1.0 for 20 lx_Con birds for each gene. Different lower-case letters above the bars denote significant differences (*p* < 0.05) among groups.

**Figure 6 animals-14-00557-f006:**
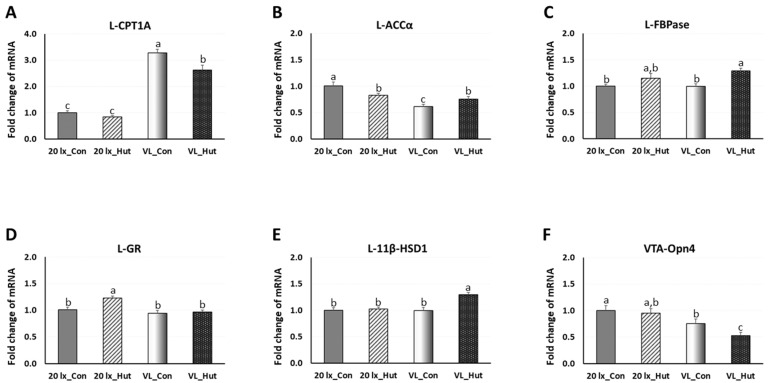
Expression changes of hepatic (**A**) L-CPT1A (liver carnitine palmitoyl transferase 1), (**B**) L-ACCα (liver acetyl-CoA carboxylase alpha), (**C**) L-FBPase (liver Fructose -1,6-bisphosphatase), (**D**) L-GR (liver glucocorticoid receptor), and (**E**) L-11β-HSD1 (liver 11 β-hydroxysteroid dehydrogenease-1) genes and (**F**) VTA Opn4 gene. Total RNA was extracted from liver and macro dissected VTA tissues and used for real time RT-qPCR. Data were set as the relative fold changes of expression levels using the ΔΔCt method with GAPDH, β-actin, and 18S as internal controls. Data (mean ± SEM) were expressed from a value set for 1.0 for 20 lx_Con birds at 14 days of age. Different lower-case letters above the bars denote significant differences (*p* < 0.05) among groups.

**Table 1 animals-14-00557-t001:** Primers used for RT-QPCR.

Gene	GenBank #	Primer Sequences (5′-3′)	Size (bp)	Annealing Tm (°C)
TPH2	NM_001001301.1	F: AGGACCTCCGCAGTGATCTA R: CAGCATAAGCAGCTGACAACA	111	58
TH	NM_204805	F: CTTTGATCCTGATGCTGCTG R: CCTCAGCTTGTTTTTGGCAT	103	56
GR	NM_001037826	F: GCCATCGTGAAAAGAGAAGG R: TTTCAACCACATCGTGCAT	95	54
BDNF	NM_001031616	F: GACATGGCAGCTTGGCTTAC R: GTTTTCCTCACTGGGCTGGA	167	60
CPT1A	NM_001012898.1	F: GTGGCTGATGATGGTTACGGT R: CCCATGATGTCAACCAATGCT	146	58
ACCα	NM_205505.1	F: GTTGCCATGGATTCGATCGTG R: GGAGTACAGGAAATCGATGCT	128	58
FBPase	AJ276212	F: TTCCATTGGGACCATATTTGG R: ACCCGCTGCCACAAGATTAC	100	58
11β-HSD1	XM_417988	F: CTGGGAACTGTCTGCACAAC R: GATTGCGAGGAACCATTTACAG	96	56
Opn4	AY036061	F: AAGGTTTCGCTGTCATCCAGC R: CTGCTGCTGTTCAAACCAAC	128	58
GAPDH	NM_204305	F: CTTTGGCATTGTGGAGGGTC R: ACGCTGGGATGATGTTCTGG	128	58–60
β-actin	L08165	F: CACAATGTACCCTGGCATTG R: ACATCTGCTGGAAGGTGGAC	158	54–56
18S	AF173612	F: TCCCCTCCCGTTACTTGGAT R: GCGCTCGTCGGCATGTA	60	60

## Data Availability

Data are unavailable due to privacy restrictions.

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
