# Peer review of "Effects of Environmental Enrichments on Welfare and Hepatic Metabolic Regulation of Broiler Chickens"

_animals, 2024, doi:10.3390/ani14040557_

Round 1

Reviewer 1 Report

Comments and Suggestions for Authors

This study proves that in broiler houses, enhanced light and fortified huts are beneficial to the mental health and liver metabolism of broilers. It is a major achievement in animal welfare.

The experiments are well designed and executed. I recommend this for the publication after the authors have addressed the following.

1. Please give a brief explanation of "environmental enrichment".

2. Line 24: "EE" to "EH"

3. Line 28: "the highest engagment" is what I want to express the highest and I don't understand.

4. Line 29: "p" needs to be changed to italics, and the full text should be checked for modification.

5. Line 157: Replace "meters" with "m"

6. Line 195: Replace "two" with "2"

7. Please change Table 1 to a three-line table

8. The "T" of "This" on line 528 does not need to be bolded

9. Line 540: Delete the second "1"

10. Line 619: Delete second "39"

Author Response

Response to comments

Thank you very much for your valuable comments. Changed words, numbers, and sentences were marked in yellow color.

  1. Please give a brief explanation of "environmental enrichment".

         Response:  Described in lines 68-72 of the revised manuscript.

  1. Line 24: "EE" to "EH"

         Response: Changed line 29 of the revised manuscript.

  1. Line 28: "the highest engagement" is what I want to express the highest and I don't understand.

         Response: Agreed. Sentence was rephrased to make clear in lines 32-33 of the revised manuscript.

  1. Line 29: "p" needs to be changed to italics, and the full text should be checked for modification.

         Response: “p” was changed to italics in all full text (yellow marked).

  1. Line 157: Replace "meters" with "m"   

         Response: “meters” was replaced in line 165 of the revised manuscript.

  1. Line 195: Replace "two" with "2"

         Response: “two” was replaced with “2” in line 173 of the revised manuscript.

  1. Please change Table 1 to a three-line table

         Response: Table 1 format was changed into a three-line table between lines 397 and 398 of the revised manuscript.

  1. The "T" of "This" on line 528 does not need to be bolded

         Response: “T” was changed into “T” in line 534 of the revised manuscript.

  1. Line 540: Delete the second "1"

      Response: Volume, and first page numbers were corrected into “11, 1281” in line 555 of the revised manuscript.

  1. Line 619: Delete second "39"

      Response: Second reference number “39” was changed to “41” and reference numbers “40-65” were change to “42-67” in the revised manuscript. There was one additional reference ([38]) added in line 145.  

Reviewer 2 Report

Comments and Suggestions for Authors

The paper presented to me for evaluation entitled" Effects of environmental enrichments on welfare, and hepatic 2 metabolic regulation of broiler chickens" addresses important issues of bird welfare in commercial farming.

However, it requires some corrections, which I list below:

The abstract is too long, please shorten it

Material and methods:

Please revise the citation in section 2.1.1 and bring it in line with the journal's requirements.

Notation 20 L: 4 D- please insert spaces.

Please indicate the density of birds included in the experiment. This is key information.

Table 1 needs to be adapted to journal requirements 

Which test for normality of distribution was used? Please indicate

Please correct the description of figure 1. It is not necessary to explain the fact that the groups labeled ab are not different from either a or b, similarly in all other cases

The discussion is written very well, but in my opinion it omits behavioral issues, or they are described very modestly. I encourage more emphasis on behavioral issues. In addition, perhaps it is worth raising the issues of the bird brain as a whole in the context of the changes the breeding process has led to? For example, I cite the work of : https://doi.org/10.1038/s41598-023-27517-3

Author Response

Response to comments

Thank you very much for your valuable comments. Changed words, numbers, and sentences were marked in yellow color.

The abstract is too long, please shorten it

Response: Number of words of abstract was reduced from 468 to 396 words in the revised manuscript.

Material and methods:

Please revise the citation in section 2.1.1 and bring it in line with the journal's requirements.

Response: reference [38] was added in line 145 and in lines 636-637 of the revised manuscript.

Notation 20 L: 4 D- please insert spaces.

Response: To make clear light program, “23 h light(L):1h dark(D) (23L:1D)” “20L:4D” and “16L:8D” were used in lines 145-147 of the revised manuscript.

Please indicate the density of birds included in the experiment. This is key information.

Response: “stocking density_12.3 birds/m2 “was added in line 136 of the revised manuscript.

Table 1 needs to be adapted to journal requirements 

Response: Tabel 1 was changed to “a three-line table” format in the revised manuscript.

Which test for normality of distribution was used? Please indicate

Response:” by Shapiro-Wilk test” was indicated in line 235 of the revised manuscript.

Please correct the description of figure 1. It is not necessary to explain the fact that the groups labeled ab are not different from either a or b, similarly in all other cases

Response: The description of figures commented by reviewer was corrected in the revised manuscript.

The discussion is written very well, but in my opinion it omits behavioral issues, or they are described very modestly. I encourage more emphasis on behavioral issues. In addition, perhaps it is worth raising the issues of the bird brain as a whole in the context of the changes the breeding process has led to? For example, I cite the work of : https://doi.org/10.1038/s41598-023-27517-3

Response: More discussion for behavioral issues and the suggested bird brain issue was described in lines 413-414, 430-436, and 441-446 of the revised manuscript based on the presenting information.